# Age Rather Than Supplementation with Oat β-Glucan Influences Development of the Intestinal Microbiota and SCFA Concentrations in Suckling Piglets

**DOI:** 10.3390/ani13081349

**Published:** 2023-04-14

**Authors:** Lidija Arapovic, Yi Huang, Elin Manell, Else Verbeek, Linda Keeling, Li Sun, Rikard Landberg, Torbjörn Lundh, Jan Erik Lindberg, Johan Dicksved

**Affiliations:** 1Department of Animal Nutrition and Management, Swedish University of Agricultural Sciences, SE-750 07 Uppsala, Sweden; 2Department of Animal Science and Technology, Guangxi University, Nanning 530004, China; 3Department of Clinical Sciences, Swedish University of Agricultural Sciences, SE-750 07 Uppsala, Sweden; 4Department of Animal Environment and Health, Swedish University of Agricultural Sciences, SE-750 07 Uppsala, Sweden; 5Department of Molecular Sciences, Swedish University of Agricultural Sciences, SE-750 07 Uppsala, Sweden; 6Department of Biology and Biological Engineering, Division of Food and Nutrition Science, Chalmers University of Technology, SE-412 96 Gothenburg, Sweden

**Keywords:** β-glucan, microbiota, SCFA, suckling piglets

## Abstract

**Simple Summary:**

Early development of the intestinal microbiota is considered critical for enteric health in pigs. Dietary strategies that improve gut microbiota development can thus help prevent disturbances associated with weaning. This study evaluated the effects of a soluble oat β-glucan supplement, compared with a placebo, in five litters of piglets during the suckling period. The supplement was introduced at week 1 of age and was provided three times per week until weaning. The effects of the supplement were evaluated in terms of weight development, concentrations of short-chain fatty acids in plasma and intestinal contents, and microbiota composition in rectal swabs and intestinal contents. The results showed that microbiota composition and short-chain fatty acid concentration changed with piglet age, but with no clear effects of the β-glucan supplement. There were no differences in body weight gain between β-glucan-supplemented and control piglets. Several correlations between specific microbes and short-chain fatty acids were identified. Significant litter effects were also found, confirming the importance of genetics and pen environment in suckling piglets.

**Abstract:**

The effects of early supplementation with oat β-glucan during the suckling period on piglet gut microbiota composition, concentrations of short-chain fatty acids, and gut physiological markers were assessed. Fifty piglets from five litters, balanced for sex and birth weight, were divided within litters into two treatment groups: β-glucan and control. Piglets in the β-glucan group received the supplement three times/week from day 7 of age until weaning. Rectal swab samples were collected from 10 piglets per treatment group (balanced across litters) from week 1 to week 4, and plasma samples were collected at 1, 3, and 4 weeks of age. Additional samples of intestinal tissues and jugular and portal vein plasma were collected from 10 animals at weaning (one per treatment group and litter). The concentrations of short-chain fatty acids in plasma and the microbiota composition in rectal swabs were mainly influenced by piglet age, rather than the supplement. There were significant differences in microbiota composition between litters and several correlations between concentrations of short-chain fatty acids in plasma and specific microbial taxa in rectal swabs. Overall, β-glucan supplementation did not have any clear impact on the gut environment in suckling piglets, whereas a clear age-related pattern emerged.

## 1. Introduction

A range of factors influence the composition of microbiota in the gastrointestinal tract of piglets, including genetics, housing conditions, use of medication and diet [1,2]. The weaning period is known to be a very stressful time in the life of piglets, because weaning is associated with abrupt separation from the sow, major dietary changes, and mixing with unfamiliar piglets. Stress can lead to increased incidence of enteric infections and intestinal inflammation [3,4]. Overgrowth of harmful bacteria is a major risk factor in the development of enteric diseases such as post-weaning diarrhoea, which can lead to financial losses due to decreased growth rate, mortality, and the extra cost of medication. Although the use of antibiotics as growth promotors has been banned in the European Union for almost two decades, antibiotics are still extensively used for therapeutic purposes, e.g., to treat gastrointestinal infections. Antimicrobial use and associated antimicrobial resistance can affect animal and also human health [5]. Thus, it is important to find alternatives to antibiotics to promote enteric health in piglets. One option is diet manipulation, targeting fibre content and quality, to stimulate microbial fermentation, feed intake, and weight gain [6]. Dietary fibre is used as a biological response modifier because it has beneficial effects on the intestinal tract and stimulates the growth of beneficial bacteria. Bacterial fermentation in the small and large intestines produces metabolites such as short-chain fatty acids (SCFA) that have been shown to have positive effects on energy metabolism, intestinal barrier, and immune system in both animals and humans [6,7,8].

Dietary fibre, in particular oat bran, contains large amounts of β-glucans, which are almost fully fermented by bacteria in the large intestine [9]. Fermentation of β-glucans has been shown to increase the concentrations of lactic acid and SCFA, reduce intestinal ammonia production, and stimulate growth of a diverse, beneficial intestinal microbiota in adult pigs [10,11]. β-glucans of microbial origin have also been shown to have immuno-stimulatory and anti-inflammatory properties and may promote survival in experimental animals infected with different microbes [12]. Fermentation of β-glucans is believed to increase the production of butyrate [10] and may be beneficial for intestinal development in young piglets because it provides an energy source for colonocytes and stimulates mucus production [13]. Butyrate has also been shown to lower the number of harmful enterobacteria and reduce inflammation in the gut [14]. Therefore, concentrations of faecal SCFA are considered a marker of gut health status [15]. However, a large fraction of the SCFA produced in the gut is absorbed, so faecal levels of SCFA do not necessarily reflect the amount of SCFA produced in the gut [16]. To obtain more comprehensive data, SCFA excreted in faeces and SCFA absorbed and circulating in blood should both be analysed. In piglets, only a few studies have measured both faecal and circulating levels of SCFA [17,18]. SCFA from plasma appears to be linked to metabolic health and is considered a good biomarker of the effects of prebiotic interventions [19]. Even fewer studies have examined prebiotic fibre supplements and their influence on circulating levels of SCFAs during the suckling period of piglets. Previous studies in piglets have mainly focused on the post-weaning period, with few studies focusing on the early-life establishment of the gut microbiota, its contribution to individual piglet performance, and whether or not it can be modified by nutritional treatments [20,21].

The aim of this study was to evaluate growth performance, gut microbiota composition, SCFA concentration in faeces and blood, and histological development of the gut in suckling piglets provided with an oat β-glucan supplement. The starting hypothesis was that the β-glucan supplementation leads to earlier development of a mature microbiota and higher SCFA production.

## 2. Materials and Methods

### 2.1. Ethics Statement

This study was approved by the Ethics Committee for the Uppsala region, with reference number DNR C 1054/16. The study was conducted in accordance with the ARRIVE guidelines [22].

### 2.2. Animals

The experiment involved 50 Hampshire x Yorkshire pathogen-free piglets from five litters [23]. Ten piglets from each litter were allocated to two equal-sized treatment groups that were balanced for sex and birth weight. In litters with more than 10 piglets, the remaining piglets were allowed to stay with the litter but were not included in the study. These selection criteria were set before the experiment.

### 2.3. Housing and Management

The experiment was carried out at the Swedish Livestock Research Centre at Lövsta, Uppsala. Pregnant sows were moved to the farrowing unit one week before expected farrowing and stayed with their piglets until weaning at 35 days of age. The farrowing pens (3.35 × 2.0 m) consisted of a heated concrete floor as the lying and feeding area (2.1 m × 2. 0 m), a slatted dung area (1.25 m × 2.0 m), and a heated corner that was only accessible to the piglets. The sows were given 15–20 kg of chopped straw two days prior to the expected farrowing date. An additional small amount of straw (0.5–1 kg/day) was given daily as enrichment after the pens were manually cleaned. Piglets were weighed, marked with an individual number within one day after birth (ear-tattoo), and ear-tagged with their individual number at five days of age. A 1-mL intramuscular injection of an iron supplement (Uniferon, 200 mg/mL) was given at five days and again at two weeks of age. All piglets had ad libitum access to creep feed (Gottfrid, Lantmännen, Sweden, Appendix A) and water during the suckling period. At weaning, the sow was removed, and the piglets remained in their pen until nine weeks of age, when they were handled according to regular routines at the Research Centre.

### 2.4. Treatments

Five suckling piglets per litter received an oral supplement of oat β-glucan (40 mg/kg body weight) (BG group) and five piglets per litter received an oral supplement of water (CON group). Disposable syringes were prefilled with a paste form of β-glucan dissolved in water, which was supplemented directly into the mouth of each pig in the BG group. The piglets were assigned to the treatment groups prior to supplementation, which started at seven days of age and was continued three times per week until weaning at five weeks of age. The pigs in the CON group were handled in the same way but received water by syringe, instead of the BG supplement. The dietary supplement used in the experiment was SweOat bran BG28 (Swedish oat fibre, Gothenburg, Sweden), which contained 28% soluble β-glucan with molecular weight 2000 kDa. The total dietary fibre content of BG28 was 52 g/100 g.

### 2.5. Experimental Procedures

An overview of the experimental set-up is shown in Figure 1. Weight gain was recorded for all piglets on a weekly basis from birth until nine weeks of age, with an extra weight measurement on the day of weaning. Four piglets per litter, two from each treatment, balanced for sex and weight between treatments, were selected for collection of faecal microbiota samples using rectal swabs (E-Swabs, Copan Diagnostics) at 1, 2, 3, and 4 weeks of age. The swabs were immediately placed on ice after sampling and within 2 h were transferred to a freezer and stored at −80 °C until analysis. Blood samples from the jugular vein of the same piglets were collected at 1, 3, and 4 weeks of age, using a 21G vacutainer needle and 4 mL VenoSafe Vacutainer EDTA-tubes. After centrifugation for 20 min at 4 °C and 1500× *g*, plasma was collected and stored at −80 °C until analysis. The oral supplementation and sampling were performed consecutively. Due to obvious differences in the characteristics of the supplements, the individual collecting the samples was not blind to the treatment. However, the individual performing the laboratory analyses was blind to the treatments, reducing any potential bias in outcome for the two treatment groups. Piglets that needed treatment with antimicrobial drugs were excluded from the study.

### 2.6. Post-Mortem Sampling Procedures

On the day before weaning, one piglet per treatment group and litter was euthanized for collection of blood and intestinal samples as described below, while the remaining piglets were weaned and handled according to existing farm routines. The five BG and five CON piglets used for post-mortem sampling were selected so that sex and body weight were balanced between the two diet treatments. The selected piglets were anaesthetized with a combination of medetomidine (Domitor^®^ vet, 1 mg/mL; Orion Pharma Animal Health, Espoo, Finland) at a dose of 0.05 mg/kg BW, and tiletamine and zolazepam (Zoletil^®^ 50 mg + 50 mg/mL; Virbac, Carros, France) at a dose of 2.5 + 2.5 mg/kg BW injected intramuscularly (i.m.). Buprenorphine (Vetergesic^®^ vet, 0.3 mg/mL; Orion Pharma Animal Health, Espoo, Finland) was given at a dose of 0.01 mg/kg BW i.m. for additional analgesia and a 22G cannula (BD^TM^ Venflon; BD, Franklin Lakes, NJ, USA) was placed in the auricular vein. Throughout the sampling procedure, anaesthesia was maintained with intravenous (i.v.) bolus doses of the pharmaceuticals listed above. Blood samples were collected from the jugular vein with a 20G vacutainer needle into EDTA vacutainer tubes. For blood collection from the portal vein, the abdomen was opened with an incision in linea alba and along the last rib. The portal vein was reached by blunt dissection, and blood was collected with a 20G needle and a 10 cc syringe and immediately transferred to an EDTA vacutainer tube. At the end of the procedure, the pigs were euthanized with pentobarbital sodium (Allfatal vet, 100 mg/mL; Omnidea, Stockholm, Sweden) i.v. while still under general anaesthesia. The blood samples were kept on ice and centrifuged and stored as described above. After euthanasia, the gastrointestinal tract was removed. A vertical incision was made immediately on the distal part of the ileum (approximately 15 cm from the caecum) and the central part of the colon, for the collection of intestinal samples. Intestinal contents collected from the colon were snap-frozen in liquid nitrogen and stored at −80 °C until analysis. In addition, one segment of tissue each from the distal ileum and central colon were taken and fixed in 10% formalin. All samples were individually re-coded and labelled before analysis, to avoid any bias in data analysis.

### 2.7. Histology of Intestinal Tissue

Ileal and colon tissue were stored in formalin with 10% neutral phosphate buffer (Merck KGaA, Darmstadt, Germany) for 48 h. After fixation, three samples per tissue and piglet were trimmed, dehydrated, cleaned, and embedded in paraffin. The paraffin-embedded tissue blocks were then cut into 4 mm thick sections, which were placed on a slide and stained with haematoxylin and eosin. A light microscope (Nikon ECLIPSE 80i, BergmanLabora AB, Danderyd, Sweden), equipped with image analysis software (NIS-Elements D 5.20.02, Nikon Instruments, Melville, NY, USA) was used for examination of gut thickness, mucosal thickness, villus height, crypt depth, and thickness of muscularis externa.

### 2.8. DNA Extraction, PCR Conditions and Sequence Analysis

DNA was extracted from rectal swabs and intestinal contents using QIAamp DNA Stool Minikit (Qiagen, Hilden, Germany) according to the manufacturer’s instructions, but with the addition of an extra mechanical lysis step using 0.1 mm zirconium/silica beads (Biospec Products, Bartlesville, OK, USA) and 2 × 1 min at 6000 rpm with a Precellys evolution (Bertin Instruments, Montigny-le-Bretonneaux, France). The isolated DNA was stored at −20 °C until analysis. PCR amplicons were generated from 16S rRNA genes using the primers 341F (CCTACGGGNGGCWGCAG) and 805R (GACTACHVGGGTATCTAATCC). PCR amplification used DreamTaq PCR chemistry with the following PCR conditions: initial denaturation at 94 °C for 3 min, followed by 25 cycles with 94 °C for 40 s, 58 °C for 40 s, and 72 °C for 60 s, and finally elongation at 72 °C for 7 min. Positive PCR reactions were confirmed with gel electrophoresis (1% agarose) using a Thermo Fisher Scientific Gene Ruler 1 kB DNA ladder as a size marker. The PCR product was purified using Agencourt AMPure purification (Beckmann Coulter, Indianapolis, IN, USA). A second PCR step was then applied to attach sample-specific barcodes and adaptor sequences. Cycling conditions were as follows: initial denaturation at 94 °C for 3 min, 8 cycles of 94 °C for 40 s, 58 °C for 40 s, and 72 °C for 60 s. PCR amplification ended with a final elongation step at 72 °C for 7 min. After a second cleaning step, all samples were quantified using a Qubit Fluorometer (Thermo Fisher Scientific, Waltham, MA, USA). Samples were pooled in equimolar amounts and sent for sequencing at SciLifeLab in Stockholm, Sweden, using Illumina MiSeq.

The sequencing data generated were processed according to the procedure described by Müller et al. [24]. The Quantitative Insights into Microbial Ecology (QIIME) version 1.8.0 pipeline was used for the quality filtering of data and to generate operational taxonomic units (OTUs) using the open reference OTU picking strategy at a threshold of 97%, with U-CLUST against a Greengenes core set (gg_13_8) [25]. Representative sequences were aligned against the Greengenes core set using PyNAST software (chimeric sequences were removed by ChimeraSlayer [26]). Taxonomy was assigned to each OTU using the Ribosomal Database Project (RDP) classifier with a minimum confidence threshold of 80% [27]. The final OTU table was filtered based on two criteria: an OTU had to be observed in three samples to be retained and an OTU had to contain 87 reads (0.001% of total reads). The sequence dataset was normalised so that each sample contained 14,900 sequences prior to analysis of microbiota. The raw sequencing data have been deposited at the National Center for Biotechnology Information database (NCBI), under accession number PRJNA940420.

### 2.9. Determination of SCFAs in Plasma and Digesta

Concentrations of SCFAs and caproic acid in EDTA plasma and digesta were analysed by liquid chromatography-mass spectrometry (LC-MS) according to a method described previously [28], with some modifications (for details, see Appendix A).

### 2.10. Statistical Analysis

Microbiota composition was analysed using both univariate and multivariate approaches. All multivariate statistical analyses were performed using the statistical software PAST, version 4.04 [29]. A multivariate approach involving principal coordinate analysis (PCoA) based on Bray–Curtis distance was performed on OTU data. Clustering patterns identified in the PCoA were validated by analysis of similarity (ANOSIM) based on 9999 permutations.

Relative abundance of the five most abundant phyla and the 10 most abundant genera, alpha diversity index values, plasma concentrations of SCFA and caproic acid, and body weight were analysed in r software (version 4.0.2.). Model assumptions of normality and homoscedasticity of the data were visually checked by QQ-plots (LMERConvenienceFunctions package) [30]. Relative abundance, alpha diversity, SCFA, caproic acid, and weight data were analysed by mixed models (packages lme4 [31] and lmerTest [32]), with treatment, sex, age, and their interactions as fixed effects and with piglet ID nested within sow ID as a random effect. If the data did not meet the normality assumptions, data transformation was applied. Data on acetate and butyrate concentrations and on phylum Proteobacteria remained untransformed, but data on valeric acid, Fusobacteria, S24_7:g, and [*Prevotella*] were square-root transformed, and all other variables were log-transformed. If the variable still did not meet normality assumption, any outliers with residuals larger than 2.5 units were excluded from the analysis (Appendix A). After data transformation and exclusion of the outliers, all variables met the normality assumption.

Stepwise backward selection was used for model selection. Post-hoc Tukey HSD tests were performed using the emmeans package (R Foundation for Statistical Computing, Vienna, Austria) [33]. Correlations between the top 10 genera and plasma concentrations of SCFA and caproic acid were analysed with linear correlations (Pearson) on samples from animals for which data on both microbiota and SCFA were available. Effects of treatment on samples collected from the euthanized animals (tissues, jugular, and portal vein plasma) were analysed with *t*-tests. To correct for multiple analyses in the correlation analysis, the Benjamini–Hochberg procedure was used with a false discovery rate (FDR) of 5% [34].

## 3. Results

### 3.1. Animal Growth and Performance

Mean birth weight of the piglets was 1.85 ± 0.04 kg (BG) and 1.86 ± 0.04 kg (CON). The animals grew well in both treatment groups, and there were no differences in growth (Figure 2) or in average daily weight gain (Appendix A) between BG and CON piglets. The piglets were healthy during the experiment, none died, and there were no cases of diarrhoea that needed treatment, although loose stools were prevalent within both groups during the experimental period.

### 3.2. Development of the Microbiota

Analysis of microbial diversity revealed a significant increase with age in both Shannon index and Chao-1 index, but with no differences between BG and CON piglets (Table 1). Shannon index was influenced by a sex by age interaction (F_(3,52.5)_ = 9.3, *p* < 0.001), with female piglets showing higher microbiota diversity than males at 4 weeks of age (*p* < 0.001). Chao-1 index was also influenced by a sex by age interaction (F _(3,54)_ = 3.8, *p* < 0.05), with females having higher species richness than males at 4 weeks of age (*p* < 0.05).

At phylum level, the microbiota was primarily dominated by Firmicutes, Bacteroidetes, and Proteobacteria (Figure 3A). At genus level, *Escherichia*, *Bacteroides*, *Prevotella*, and *Lactobacillus* were most dominant, comprising more than 50% of relative abundance in swab samples (Figure 3B).

To further assess microbial community composition, a multivariate statistical approach was applied. The ordination with PCoA revealed a clustering pattern that was linked to piglet age but did not indicate any difference in clustering pattern between BG and CON pigs (Figure 4A). This was confirmed in two-way ANOSIM, where age was a significant factor (*p* = 0.0001, R = 0.24) but not treatment (*p* = 0.32, R = 0.013). The PCoA and ANOSIM results were also used to evaluate the effect of litter on microbiota composition and revealed an evident effect of litter (Figure 4B), with a significant difference between both age and litter in the two-way ANOSIM (age: *p* = 0.0001, R = 0.43; litter: *p* = 0.0001, R = 0.45).

A univariate statistical approach was also applied to evaluate the effect of treatment and age on the five most abundant phyla and the 10 most abundant genera (Table 2). At phylum level, there was a tendency for a treatment x age interaction for Actinobacteria (F_(3,53.4)_ = 2.5, *p* = 0.07), with BG piglets tending to have higher relative abundance than CON piglets at week 1 (*p* = 0.07) (Table 2). Relative abundance of Actinobacteria decreased with age (F_(3,53.4)_ = 8.24, *p* < 0.001) (Table 2). Males (0.9 ± 0.2%) had a higher relative abundance of Actinobacteria than females (0.4 ± 0.1%), F_(1,13.3)_ = 5.9, *p* < 0.05) (Table 2). A tendency for a treatment x age interaction was found for Bacteroidetes (F_(3.68)_ =2.4, *p* = 0.07), with higher relative abundance at week 1 in pigs from the BG group (*p* = 0.007) (Table 2). Relative abundance of Firmicutes was also influenced by age (F_(3,72.0)_ = 5.1, *p* < 0.01, with a significant decrease in abundance at week 4 compared with week 2 (*p* < 0.05), and week 3 compared with week 4 (*p* < 0.01) (Table 2). Moreover, Fusobacteria was influenced by age (F_(3,55,9_) = 34.4, *p* < 0.0001, with abundance decreasing significantly from week 1 to week 2 (*p* < 0.001), week 3 (*p* < 0.001) and week 4 (*p* < 0.001) (Table 2). There was a tendency for a sex effect on Proteobacteria (F_(1,14.3)_ = 4.2, *p* = 0.06), where male piglets had a higher abundance of Proteobacteria than females (Table 2).

At genus level, relative abundance of *Escherichia* was influenced by a treatment x age interaction (F_(3,63)_ = 3.5, *p* < 0.05), with higher relative abundance in the BG group at 4 weeks of age (*p* = 0.003) (Table 2). An age x sex interaction was also found (F_(3,63)_ = 4.7, *p* < 0.01), where female piglets had higher relative abundance of *Escherichia* than male piglets at week 4 (*p* < 0.001). Relative abundance of *Bacteroides* was influenced by a treatment x age interaction (F_(3,62)_ = 3.6, *p* < 0.05), with higher abundance in the BG group in week 1 (*p* < 0.05), but higher abundance in the CON group in week 2 (*p* < 0.05). In addition, an age × sex interaction was found (F_(3,62)_ = 4.0, *p* < 0.05), with higher relative abundance in male piglets compared with females (*p* < 0.01) (Table 2). Relative abundance of *Prevotella* was influenced by a treatment x sex x age interaction (F_(3,46.5)_ = 6.6, *p* < 0.001), where in week 1 (*p* < 0.01), week 2 (*p* < 0.05), and week 3 (*p* < 0.05), female BG piglets had a higher relative abundance of *Prevotella* than female CON piglets (Table 2). Relative abundance of *Lactobacillus* was influenced by age (F_(3,56)_ = 18.6, *p* < 0.001) and was higher in weeks 1 and 2 compared with weeks 3 and 4 (*p* < 0.001) (Table 2). The relative abundance of *Ruminococcus* was influenced by age (F_(3,72.2)_ = 8.9, *p* < 0.001), with abundance increasing from week 1 to week 2 (*p* = 0.001) and decreasing from week 2 to week 4 (*p* < 0.001) (Table 2). The relative abundance of *Oscillospira* was influenced by age (F_(3,71.9)_ = 6.8, *p* < 0.001), with lower abundance in week 1 compared with week 2 (*p* = <0.001), week 3 (*p* < 0.05), and week 4 (*p* < 0.05) (Table 2). The abundance of S24_7:g was influenced by an age x sex interaction (F_(3,52.9)_ = 3.3, *p* < 0.05), where female piglets had higher abundance at week 4 (*p* < 0.05) than male piglets (Table 2). Relative abundance of [*Prevotella*] was influenced by age (F_(3,56.7)_ = 8.5, *p* < 0.001), with a higher abundance at week 3 (*p* < 0.05) and week 4 (*p* < 0.001) compared with week 1, and a higher abundance in week 4 (*p* < 0.05) compared with week 2 (Table 2). *Ruminococcus* was influenced by age (F_(3,72.2)_ = 8.9, *p* < 0.001), with abundance decreasing from week 1 to week 2 (*p* < 0.001) and week 3 (*p* < 0.05), and from week 2 to week 4 (*p* < 0.05) (Table 2).

The microbiota in colon digesta from euthanized piglets from the BG and CON groups were also compared, but PCoA did not reveal any separate clustering of these samples according to treatment or any significant effects of the supplement (Appendix A).

### 3.3. SCFA and Caproic Acid Concentrations in Plasma and Colon Digesta

Analysis of SCFA and caproic acid levels in plasma samples collected from the jugular vein at 1, 3, and 4 weeks of age revealed that the concentration of SCFA was mainly affected by age. The only observed effect of treatment was on acetic acid concentration, which was lower in the BG group than in the CON group (F_(1,18)_ = 6.1, *p* < 0.05) (Table 3). Acetic acid concentration was also influenced by age (F_(2,38)_ = 39.8, *p* < 0.001), with an increase from week 1 to week 3 (*p* < 0.001) and week 4 (*p* < 0.001). The concentration of propionic acid was influenced by age (F_(2,38)_ = 3.3, *p* < 0.05), with the highest concentration at week 3 (*p* < 0.05) (Table 3). Plasma concentration of butyric acid was influenced by an age x sex interaction (F_(2,34.6)_ = 4.5, *p* < 0.05), with higher concentrations in females than in males in week 4 (*p* < 0.05). Formic acid concentration was influenced by age (F_(2,38)_ = 25.7, *p* < 0.001), decreasing from week 1 to week 3 (*p* < 0.01) and week 4 (*p* < 0.001), and from week 3 to week 4 (*p* < 0.01) (Table 3). Iso-butyric acid concentration was influenced by age (F_(2,38)_ = 6.3, *p* < 0.05), and was higher in week 3 (*p* < 0.05) and week 4 (*p* < 0.05) than in week 1 (Table 3). Valeric acid concentration tended to be higher in the BG than in the CON group (F_(1,18)_ = 4.0, *p* = 0.06) and was influenced by age (F_(2,38)_ = 9.3, *p* < 0.001), with an increase from week 1 to week 3 (*p* < 0.05) and week 4 (*p* < 0.001). Succinic acid concentration was influenced by age (F_(2,38)_ = 5.4, *p* < 0.05) and decreased from week 1 to week 4 (*p* < 0.05) (Table 3). Caproic acid concentration tended to be higher in BG piglets than in CON piglets (F_(1,12)_ = 3.5, *p* = 0.08) (Table 3).

Jugular vein and portal vein plasma and colon digesta taken from 10 euthanized animals were evaluated for the possible influence of the BG supplement on SCFA concentrations. SCFA concentration varied between individual pigs and was considerably higher in plasma samples from the portal vein than in samples from the jugular vein. However, no significant effects linked to the treatment were found for colon digesta or plasma samples (Appendix A). The molar proportions of SCFA differed between plasma and colon digesta, with higher similarity in portal vein and colon digesta than in jugular vein and colon digesta (Figure 5). However, correlation analysis of individual SCFA in the different sample types did not reveal any significant correlations.

### 3.4. Gut Histological Measurements

Morphological parameters measured in tissue samples are shown in Table 4. No significant differences were detected between BG and CON for any of the measured parameters.

### 3.5. Correlations

Relative abundance data from the 10 most abundant genera were analysed for correlations with SCFA in plasma (Appendix A). Among the genera classified to the phylum Bacteroidetes, *Bacteroides* was negatively correlated with acetic acid (r = −0.34, *p* = 0.008) and propionic acid (r = −0.35, *p* = 0.004), while *Prevotella* was positively correlated with acetic acid (r = 0.42, *p* = 0.0008), but negatively correlated with formic acid (r = −0.31, *p* = 0.016). The taxon S24_7:g was negatively correlated with formic acid (r =−0.48, *p* = 0.0001), but positively correlated with acetic acid (r = 0.55, *p* = 0.000005), propionic acid (r = 0.34, *p* = 0.007), butyric acid (r = 0.49, *p* = 0.00005), iso-butyric acid (r = 0.34, *p* = 0.009), and valeric acid (r = 0.33, *p* = 0.01). The taxon [*Prevotella*] was positively correlated with acetic acid (r = 0.51, *p* = 0.00003), propionic acid (r = 0.37, *p* = 0.003), and butyric acid (r = 0.39, *p* = 0.002). Among the dominating genera within the phylum Firmicutes, *Lactobacillus* was positively correlated with succinic acid (r = 0.36, *p* = 0.005) but negatively correlated with acetic acid (r = −0.46, *p* = 0.0002). Lachnospiraceae;g was positively correlated with valeric acid (r = 0.31, *p* = 0.016), Ruminococcaceae;g_other was positively correlated with propionic acid (r = 0.36, *p* = 0.004) and iso-butyric acid (r = 0.31, *p* = 0.015), and *Oscillospira* was positively correlated with iso-butyric acid (r = 0.33, *p* = 0.011). Moreover, *Ruminococcus* showed a positive correlation with formic acid (r = 0.42, *p* = 0.0008) and a negative correlation with acetic acid (r = −0.39, *p* = 0.002). *Escherichia,* belonging to Proteobacteria, was positively correlated with caproic acid (r = 0.33, *p* = 0.009). There were also several correlations between individual bacterial taxa (Appendix A).

## 4. Discussion

This study investigated the effect of early dietary supplementation with β-glucan on the development of the gut microbiota, concentrations of SCFA in plasma and digesta, and the morphological structure of the gut in piglets. The results showed mainly age-related effects, with maturation of microbiota composition and higher concentrations of certain SCFA in plasma at older age. Several correlations were found between certain microbial taxa and concentrations of SCFA and caproic acid in plasma samples.

Early in the piglets’ life, *Escherichia*, *Lactobacillus*, and *Bacteroides* were dominating taxa, but with older age both *Lactobacillus* and *Bacteroides* showed a reduction in relative abundance. The relative abundance of *Lactobacillus* decreased from 9% initially to 2% by the end of the suckling period. Some previous studies have found a similar trend, with a significant decrease in the abundance of *Lactobacillus* from newly weaned piglets compared to piglets at the end of the suckling period [35,36]. In contrast, Frese et al. [37] found an increase in the proportion of *Lactobacillus* directly after weaning. In the present study, we observed a decrease in the relative abundance of *Bacteroides* with older age, which is in agreement with findings in other studies [37,38].

A gradual increase in the relative abundance of *Prevotella* was observed during the suckling period, which is in agreement with findings in earlier studies on gut microbiome sampled at different ages [38,39,40,41,42]. *Prevotella* is known to harbour specific enzymes that can utilise plant polysaccharides [43]. Thus, it is commonly present in low abundance in nursing animals and usually increases in abundance when pigs are introduced to solid food [38]. Interestingly, in the present study, *Prevotella* showed a gradual increase over the whole suckling period, possibly due to increasing intake of creep feed with age during the suckling period [44]. Consumption of creep feed has been shown to contribute to microbiota development [20,21]. Creep feed consumption was not recorded in the present study, so no conclusions can be drawn on whether it contributed to the observed increase in the abundance of *Prevotella*.

Several differences in microbial composition correlated with piglet sex were observed. Relative abundance of the phylum Proteobacteria was higher in male piglets, while at genus level *Escherichia* showed higher abundance in female piglets and *Bacteroides* showed higher abundance in male piglets. However, previous studies have found conflicting results, e.g., Zhou et al. [45] found a higher abundance of Proteobacteria in female pigs, and He et al. [46] found a higher abundance of *Escherichia* in males and *Bacteroides* in females. The reason for the discrepancies between studies is unknown, but one possible explanation is age differences, as the pigs in those studies were older than the piglets in our study.

It is uncertain to what extent the immature gut microbiota in young piglets can utilise dietary fibre [47]. Microbial metabolism of dietary fibre leads to various end-products [48], and the production of SCFA is considered to be a key factor influencing overall gut health. A substantial part of the energy contribution from the gut in weaned pigs is related to SCFA, with intestinal production of SCFA depending on the size of the gastrointestinal tract and the population and composition of bacteria in the gut [49]. In the present study, changes in SCFA concentrations in plasma were linked more strongly to age than to treatment. The only observed effect linked to the treatment was for acetic acid. However, the acetic acid concentration in plasma was higher in CON piglets at the start of the study, so this difference likely reflected individual differences rather than being an effect of the supplementation.

Formic and succinic acid concentrations in plasma decreased with age, whereas the concentrations of acetic, propionic, butyric, iso-butyric, and valeric acid all increased with age. As the abundance of *Bacteroides* and *Lactobacillus* decreased with age, while the levels of acetic and propionic concentration increased, it was not surprising that these variables were negatively correlated. In contrast, *Prevotella* increased with age and was positively correlated with concentrations of acetic, butyric, and valeric acid. Previous studies have found that *Lactobacillus* spp., *Bacteroides* spp., *Bifidobacterium* spp., *Streptococcus* spp., *Prevotella* spp., and *Ruminococcus* spp. are the most important producers of acetate in the gastrointestinal tract [50,51]. Propionate is mainly produced by *Bacteroides* spp., *Megasphaera elsdenii*, *Roseburia inulinivorans*, *Dialister* spp., *Ruminococcus obeum*, *Veillonella* spp., *Coprococcus catus*, and *Phascolarctobacterium succinatutens* [52]. In the present study, *Prevotella* aligned with prior knowledge of the specific gut bacteria producers of acetate, but *Lactobacillus* spp. and *Bacteroides* spp. were negatively correlated with the concentrations of acetic and propionic acid. This could be due to the fact that SCFA were measured in plasma and microbiota taxa in rectal swabs. It has been reported in several previous studies that the concentrations and proportions of circulating SCFA and SCFA in faeces samples are not well correlated [53,54,55]. There was a clear difference in the concentrations of SCFA and caproic acid between portal and jugular vein plasma, with higher concentrations in the portal vein samples (even higher than found in the colon digesta). The observed difference in molar proportions between the portal and jugular vein samples, with reduced proportions of both propionic and butyric acid, suggests that they were metabolised in the liver.

Previous studies on weaned pigs have found that inclusion of oat or barley β-glucan or oat products in the diet is associated with an increased proportion of specific microbial taxa in the colon and ileum, i.e., *Lactobacillus* spp., *Bifidobacterium* spp., and *Prevotella* [56,57,58]. Other studies have found no changes in the gut microbial community, plasma SCFA, or immune function on adding oat β-glucan to the diet of pigs [59,60]. In the present study, there were no significant differences in microbiota, SCFA concentrations, or gut morphological structure that could be linked to the BG treatment alone. The β-glucan supplement was introduced from seven days of age, with three doses per week until weaning. It is possible that more frequent doses or a higher dose than 40 mg/kg body weight could have resulted in an effect of the treatment on the gut microbiota, and hence increased SCFA production. Previous studies on weaned pigs receiving β-glucan from cereals and microbial origin have used doses ranging from 50 to 200 mg/kg body weight [61]. A recent study on weaned pigs receiving a dietary β-glucan supplement and challenged with *Escherichia coli* used a high dose of β-glucan (500 mg/kg BW) and observed an increased abundance of *Lactobacillus* in the supplemented group compared with the control group [62].

Apart from most previous studies being performed post-weaning, on older pigs with a more developed microbiota that can utilise β-glucan more effectively, other possible reasons for the differences in response between studies could be related to fermentation rate in the digestive tract, which depends on the chemical composition and physiochemical properties of different types of β-glucans [63,64]. β-glucans can differ widely in both concentration and solubility, depending on the source [65]. For instance, β-glucan from fungi and yeasts is known to have immuno-modulating effects [66,67]. Yeast-derived β-glucan has high bioactivity [68], and effects on the immune system have been demonstrated in vitro [69,70] and in a recent in vivo study [71]. Although barley has a higher β-glucan content than oats [72], β-glucan from oats has been shown to be better for gut health than barley β-glucan when fed to rats at the same dosage [73].

The lack of differences between the treatment groups in the present study could potentially also be related to microbial cross-contamination by allocoprophagy [74], as we provided the CON and BG supplement to individuals within the same litters. However, this study design had many advantages, e.g., siblings from the BG and CON groups had a shared mother, genetic predisposition, and early-life litter exposure. We observed clear litter effects on gut microbial community clusters in our study. The univariate statistical analyses were litter-adjusted, so these effects were accounted for in the statistical model.

Previous studies on weaned pigs fed β-glucan derived from cereal and microbial sources have reported increased levels of butyrate and butyrate-producing bacteria [56,75]. Butyrate can act as an inhibitor of pathogenic microorganisms by reinforcing the mucus layer, enhancing immunological development, and acting as a nutrient source for the colon epithelium [76,77]. In the present study, the concentration of butyric acid was numerically, but not statistically significantly, higher in plasma samples from the BG group, but not in digesta samples, indicating that BG supplementation contributed to higher concentrations of butyrate and butyrate-producing bacteria. Families such as Lachnospiraceae and Ruminococcaceae, which contain several genera known to produce butyrate, were found in the samples, but did not differ in relative abundance between the treatments.

It has been shown that transition to solid feed can cause short-term villus and crypt disorders in piglets [78]. Moreover, it is suggested that 14–21 days of life is a crucial period for maturation of the mucosal structure and mucosal development [79]. However, no measurable changes in intestinal mucosa were seen in BG piglets in our study, and we found no indications of local inflammation in villi and crypts in any of the piglets. The effects of dietary fibre on gut morphology in piglets cannot be assessed based on the literature, as some studies report changes in morphology in piglets fed solid feed before weaning [80] and others report no changes [81].

Comparison of the results from this and previous studies is challenging, due to wide variation in the data obtained, differences in supplement dose and timing, and lack of studies on β-glucan in young piglets during the suckling period.

## 5. Conclusions

Early supplementation of piglets with oat β-glucan during the suckling period had no obvious or possible long-lasting effects, with piglet age and litter rather than dietary β-glucan supplementation having the most effect on gut microbiota development and on SCFA concentrations. This suggests that cereal β-glucan has a limited capacity to contribute to a beneficial gut microbiota development during the suckling period.

## Figures and Tables

**Figure 1 animals-13-01349-f001:**
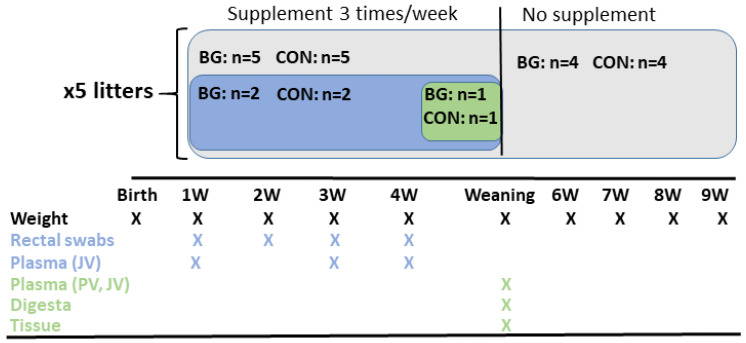
Overview of the experimental set-up. BG = β-glucan, CON = control, JV = jugular vein, PV = portal vein.

**Figure 2 animals-13-01349-f002:**
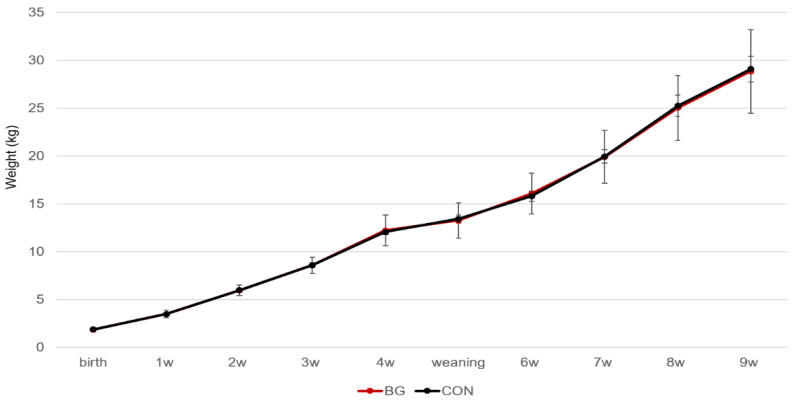
Weight gain (mean ± SE) of piglets in the β-glucan supplemented group (BG) and control group (CON) from birth to nine weeks of age.

**Figure 3 animals-13-01349-f003:**
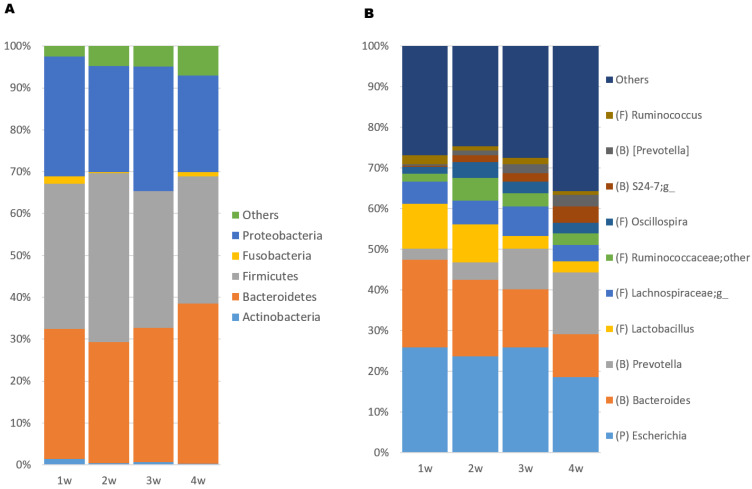
Stacked bar plot showing relative abundance of (**A**) the five most abundant bacterial phyla and (**B**) the 10 most abundant genera in rectal swab samples collected from piglets over the whole suckling period. Others = remaining accumulated abundance of bacterial taxa not among the top 10 in abundance in the samples, B = Bacteroidetes, F = Firmicutes, P = Proteobacteria.

**Figure 4 animals-13-01349-f004:**
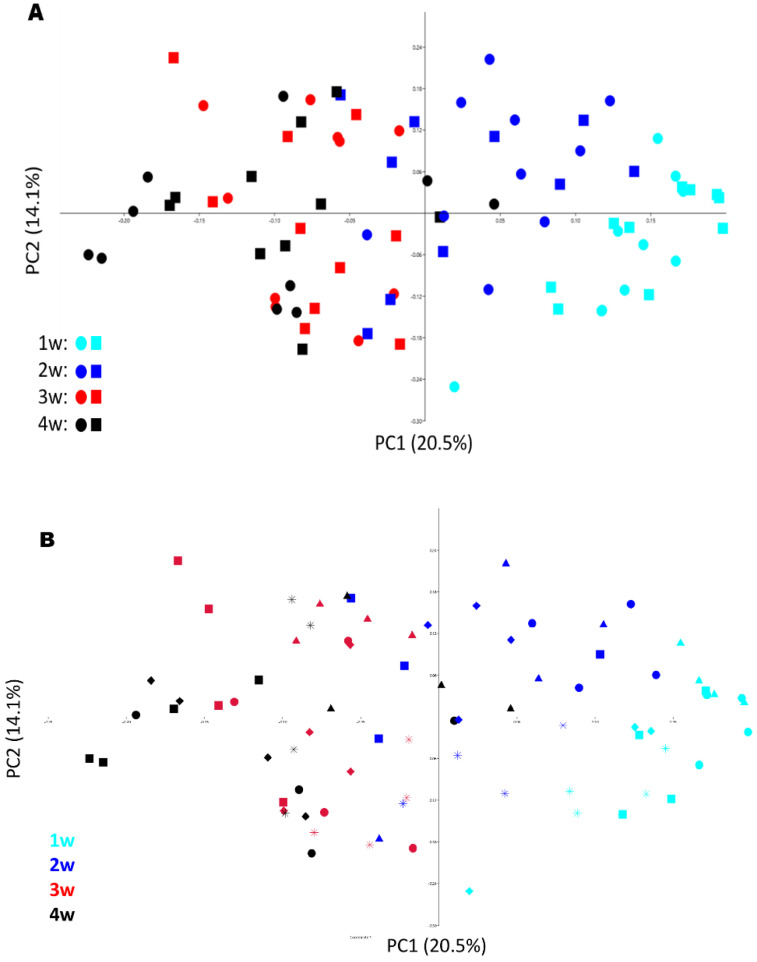
(**A**,**B**) Principal coordinates analysis (PCoA) plots with BrayCurtis distances showing microbiota development in 20 piglets (swab samples) from 1–4 weeks of age (10 receiving a β-glucan (BG) supplement and 10 from the control group (CON)). Symbols in plot (**A**) represent different treatments (circles = BG, squares = CON) and symbols (squares, diamonds, circles, triangles and stars) in plot (**B**) different litters.

**Figure 5 animals-13-01349-f005:**
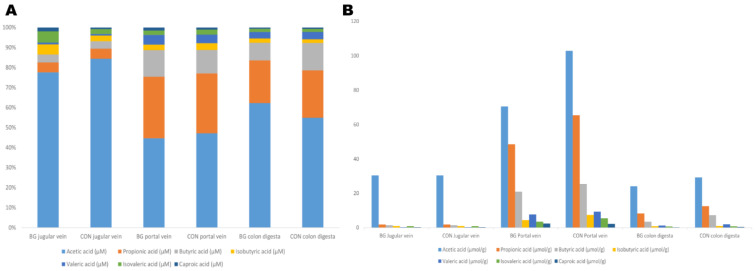
(**A**) Relative concentrations (% of total) and (**B**) mean absolute concentrations (μmol/g) of SCFA and caproic acid in jugular vein and portal vein plasma and colon digesta from 10 euthanized piglets (five piglets in a group receiving a β-glucan supplement and five control (CON) piglets.

**Table 1 animals-13-01349-t001:** Microbial diversity (mean ± SE) determined based on Illumina sequencing data in samples collected from piglets receiving a β-glucan supplement (BG) and control (CON) piglets.

		BG				CON			T	A	A:S	T:A
	1w	2w	3w	4w	1w	2w	3w	4w				
Shannon index	3.26 ± 0.13	3.59 ± 0.19	3.63 ± 0.14	4.19 ± 0.21	3.13 ± 0.17	3.68 ± 0.14	3.73 ± 0.19	4.21 ± 0.26 ^a^	ns	*	*	ns
Chao-1	538 ± 38	803 ± 35	910 ± 40	980 ± 50	535 ± 30	798 ± 39	939 ± 38	977 ± 62 ^a^	ns	*	*	ns

* indicates *p* < 0.05, ns = not significant. T = treatment, A = age, A:S = sex:age interaction, T:A = treatment:age interaction; w = weeks, ^a^ = significant difference at 4 weeks of age.

**Table 2 animals-13-01349-t002:** Relative abundance (mean percentage ± SE) on phylum and genus level of microbial taxa in swab samples collected from piglets receiving a β-glucan supplement (BG) and control (CON) piglets during the suckling period.

Bacteria	BG				CON				T	A	T:A
	1w	2w	3w	4w	1w	2w	3w	4w			
Phylum											
Actinobacteria	0.88 ± 0.40 ^a^	0.35 ± 0.08 ^b^	1.01 ± 0.35 ^b^	0.35 ± 0.13 ^b^	2.01 ± 0.66 ^a^	0.41 ± 0.19 ^b^	0.43 ± 0.15 ^b^	0.30 ± 0.19 ^b^	ns	*	ns
Bacteroidetes	40.1 ± 4.67 ^a^	26.3 ± 3.07	30.8 ± 4.47	33.7 ± 4.16	27.6 ± 6.38 ^a^	31.6 ± 3.59	33.2 ± 4.58	42.6 ± 3.79	ns	ns	ns
Firmicutes	34.6 ± 3.32	41.7 ± 3.71 ^a^	33.3 ± 2.10 ^b^	32.3 ± 3.62 ^b^	34.8 ± 3.20	39.1 ± 2.84 ^a^	32.0 ± 2.83 ^b^	28.5 ± 2.77 ^b^	ns	*	ns
Fusobacteria	1.55 ± 0.34 ^a^	0.11 ± 0.05 ^b^	0.01 ± 0.01 ^b^	0.08 ± 0.03 ^b^	1.29 ± 0.33 ^a^	0.21 ± 0.11 ^b^	0.03 ± 0.01 ^b^	0.42 ± 0.30 ^b^	ns	*	ns
Proteobacteria	22.9 ± 5.08	27.7 ± 5.23	30.1 ± 4.45	27.6 ± 5.59	34.3 ± 6.51	23.2 ± 3.26	29.4 ± 6.09	18.5 ± 4.93	ns	ns	ns
Genera											
*Escherichia*	19.9 ± 5.15	25.6 ± 5.27	30.2 ± 3.75	23.7 ± 5.79 ^A^	32.0 ± 6.46	21.6 ± 3.22	24.5 ± 5.35	13.5 ± 4.86 ^A^	ns	ns	*
*Bacteroides*	28.4 ± 4.75 ^A^	13.9 ± 3.24 ^B^	14.4 ± 3.03	10.6 ± 2.06	17.4 ± 4.27 ^A^	23.9 ± 3.57 ^B^	14.1 ± 1.97	11.4 ± 3.24	ns	ns	*
*Prevotella*	3.64 ± 1.60 ^a^	7.31 ± 2.28 ^b,c,A^	9.46 ± 3.46 ^b,d^	12.1 ± 3.70 ^b,d^	2.0 ± 0.61 ^a^	1.99 ± 0.43 ^b,c,A^	10.6 ± 3.10 ^b,d^	18.3 ± 5.65 ^b,d^	ns	*	*
*Lactobacillus*	9.44 ± 1.95 ^a^	8.45 ± 1.82 ^c^	4.65 ± 1.53 ^b,d,e^	2.63 ± 0.79 ^b,d,f^	12.5 ± 3.12 ^a^	12.1 ± 2.60 ^c^	1.67 ± 0.37 ^b,d,e^	2.95 ± 0.60 ^b,d,f^	ns	*	ns
*Lachnospiraceae*	7.73 ± 2.64	7.44 ± 2.96	7.33 ± 2.60	4.45 ± 1.67	3.10 ± 1.56	4.33 ± 1.36	7.05 ± 3.50	3.49 ± 0.99	ns	ns	ns
*Ruminococcaceae*; other	1.93 ± 0.37 ^a^	1.53 ± 0.70 ^b,c^	1.00 ± 0.15	0.97 ± 0.21 ^d^	2.39 ± 0.51 ^a^	0.63 ± 0.09 ^b,c^	2.03 ± 0.82	0.99 ± 0.36 ^d^	ns	*	ns
*Oscillospira*	1.78 ± 0.37 ^a^	3.82 ± 0.74 ^b^	2.74 ± 0.47 ^b^	3.43 ± 0.84	1.46 ± 0.45 ^a^	4.03 ± 0.64 ^b^	3.21 ± 0.79 ^b^	1.89 ± 0.28	ns	*	ns
S24_7:g	0.27 ± 0.12 ^a^	2.11 ± 0.68 ^b,c^	1.94 ± 0.41 ^b^	3.11 ± 0.91 ^b,d^	0.29 ± 0.13 ^a^	1.31 ± 0.36 ^b,c^	2.00 ± 0.48 ^b^	2.72 ± 0.55 ^b,d^	ns	*	ns
[*Prevotella*]	0.53 ± 0.44 ^a^	1.21 ± 0.48	0.88 ± 0.40 ^b,c^	1.93 ± 0.44 ^b,d^	0.29 ± 0.16 ^a^	0.94 ± 0.34	2.56 ± 1.06 ^b,c^	3.66 ± 1.14 ^b,d^	ns	*	ns
*Ruminococcus*	1.93 ± 0.37 ^a^	1.53 ± 0.70 ^b^	1.00 ± 0.15 ^c^	0.97 ± 0.21 ^d^	2.39 ± 0.51 ^a^	0.63 ± 0.09 ^b^	2.03 ± 0.82 ^c^	0.99 ± 0.36 ^d^	ns	*	ns

* Indicates *p* < 0.05, ns = not significant, T = treatment, A = age, T:A = treatment:age interaction, w = weeks, ^a,b,c,d,e,f^ = significant difference within weeks of age, ^A,B^ = significant difference within weeks of age:treatment.

**Table 3 animals-13-01349-t003:** Concentrations (µM) of different short-chain fatty acids (SCFA) and caproic acid in plasma collected from piglets receiving a β-glucan supplement (BG) and control (CON) piglets during the suckling period. Mean ± SE.

SCFA	BG			CON			T	A	T:A
	1w	3w	4w	1w	3w	4w			
Acetic acid	10.9 ± 0.69 ^a^	17.1 ± 0.85 ^b,c^	18.9 ± 1.10 ^b,d^	13.0 ± 0.76 ^a^	18.4 ± 1.44 ^b,c^	21.7 ± 1.27 ^b,d^	*	*	ns
Propionic acid	0.88 ± 0.12	1.19 ± 0.20 ^a^	1.27 ± 0.23	1.03 ± 0.09	1.82 ± 0.34 ^a^	1.21 ± 0.21	ns	*	ns
Butyric acid	0.15 ± 0.30	0.38 ± 0.71 ^a^	0.68 ± 0.10 ^a^	0.21 ± 0.05	0.34 ± 0.11 ^a^	.045 ± 0.08 ^a^	ns	*	ns
Formic acid	576 ± 20.3 ^a^	447 ± 34.7 ^b,c^	346 ± 30.1 ^b,d^	546 ± 17.6 ^a^	494 ± 53.4 ^b,c^	419 ± 50.5 ^b,d^	ns	*	ns
Iso-butyric acid	0.42 ± 0.04 ^a^	0.69 ± 0.16 ^b^	0.81 ± 0.12 ^b^	0.43 ± 0.06 ^a^	0.83 ± 0.13 ^b^	0.57 ± 0.08 ^b^	ns	*	ns
Valeric acid	0.08 ± 0.00 ^a^	0.14 ± 0.03 ^b^	0.16 ± 0.03 ^b^	0.06 ± 0.02 ^a^	0.08 ± 0.02 ^b^	0.11 ± 0.01 ^b^	ns	*	ns
Succinic acid	9.07 ± 1.66 ^a^	7.62 ± 0.94	7.28 ± 1.36 ^b^	11.89 ± 1.80 ^a^	9.34 ± 1.38	7.52 ± 0.96 ^b^	ns	*	ns
Caproic acid	0.27 ± 0.02	0.29 ± 0.02	0.24 ± 0.03	0.23 ± 0.03	0.23 ± 0.04	0.20 ± 0.03	ns	ns	ns

* Indicates *p* < 0.05, ns = not significant, T = treatment, A = age, T:A = treatment:age interaction, w = weeks, ^a,b,c,d^ = significant difference within weeks.

**Table 4 animals-13-01349-t004:** Villus height (µm), crypt depth (µm), and thickness of mucosa, total gut (µm), and muscularis externa of the ileum and colon in piglets receiving a β-glucan supplement (BG) and in control (CON) piglets during the suckling period. Mean ± SE.

	N ^(1)^	CON	BG	*p*-Value
Ileum (µm)				
Villus height	10	346 ± 19.6	360 ± 25.1	0.58
Crypt depth	5–9	234 ± 15.3	238 ± 21.9	0.82
Thickness mucosa	10	598 ± 30.7	619 ± 36.0	0.62
Thickness muscularis externa	10	543 ± 33.3	480± 24.5	0.34
Thickness total gut	5	1925 ± 70.1	2199 ± 274	0.65
Colon (µm)				
Crypt depth	10	410 ± 13.0	376 ± 17.2	0.42
Thickness mucosa	10	483 ± 14.0	445 ± 9.4	0.61
Thickness muscularis externa	10	400 ± 20.5	366 ± 20.4	0.57
Thickness total gut	5–6	1152 ± 80.0	1042 ± 76.5	0.57

^(1)^ Number of measurements per pig.

## Data Availability

The raw sequencing data have been deposited at the National Center for Biotechnology Information database (NCBI), under accession number PRJNA940420. Other data from this research study are available from the corresponding author upon request.

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
