# Peer review of "Age Rather Than Supplementation with Oat β-Glucan Influences Development of the Intestinal Microbiota and SCFA Concentrations in Suckling Piglets"

_animals, 2023, doi:10.3390/ani13081349_

Round 1

Reviewer 1 Report

The manuscript titled "Age rather than supplementation with oat β-glucan influences the development of the intestinal microbiota and SCFA concentrations in suckling piglets" described theeffects of early supplementation with oat β-glucan during the suckling period on the gut microbiota composition, concentrations of short-chain fatty acids and gut physiological markers. Authors found that supplementation with BG did not show any clear impact on the suckling piglets gut environment, whereas a clear age related pattern was seen. My main concerns are listed as follows"

(1) The language should be carefully polished by a native English speaker.

(2) Introduction: More recent and updated references should be cited here.

(3) Line 159: 341F 805R.

(4) Line 170-172: The generated sequencing data was processed according to the procedure described by Müller et al.[31]. The Quantitative Insights into Microbial Ecology (QIIME) version 1.8.0 pipeline was used for quality filtering of data and to generate operational taxonomic units (OTUs) using the open reference OTU picking strategy at a threshold of 97%. Why not using QIIME 2 and ASVs instead of OTUs?

(5) Line 183-184: The levels of SCFAs and caprionic acid were analyzed in plasma and digesta by LC-MS according to a method that has been described earlier, with some minor modifications. a method that has been described earlier? Which reference? Please provide a detailed description in this section instead of in Supplementary.

(6) An intuitive experimental design diagrams should be provided to the methos section, it is hard to catch the main design in its current way.

Author Response

We thank the reviewer for constructive feedback on the manuscript! The suggested revisions have improved the manuscript!

(1) The language should be carefully polished by a native English speaker.
Author reply: We have sent the manuscript for a professional linguistic review.

(2) Introduction: More recent and updated references should be cited here.
Author reply: We have updated the introduction with more recent references.

(3) Line 159: 341F 805R.
Author reply: Changed 

(4) Line 170-172: The generated sequencing data was processed according to the procedure described by Müller et al.[31]. The Quantitative Insights into Microbial Ecology (QIIME) version 1.8.0 pipeline was used for quality filtering of data and to generate operational taxonomic units (OTUs) using the open reference OTU picking strategy at a threshold of 97%. Why not using QIIME 2 and ASVs instead of OTUs?
Author reply: Indeed, the QIIME 2 pipeline generating ASVs is a newer pipeline for processing of amplicon sequencing data. The bioinformatics analysis in this project was done a while ago, thus the use of the older pipeline. However, in our data analysis, we have done the analysis on genus and phylum level data and on these phylogenetic levels, there are no big differences between the newer QIIME 2 and the older QIIME 1 pipelines. We have tested this in other projects with similar outcomes, thus the outcome from the data analysis in this study, will not change by re-analyzing the data using QIIME 2.  

(5) Line 183-184: The levels of SCFAs and caprionic acid were analyzed in plasma and digesta by LC-MS according to a method that has been described earlier, with some minor modifications. a method that has been described earlier? Which reference? Please provide a detailed description in this section instead of in Supplementary.
Author reply: We have updated the text with a reference to the paper where the method was described. We still keep a description of the modifications of the method in the supplement as it is quite long. This is in line how we did in earlier studies using this method.

(6) An intuitive experimental design diagrams should be provided to the methods section, it is hard to catch the main design in its current way.
Author reply: We have added a schematic illustration (new Figure 1) in the method section that describe the study design. Hope that this new figure make the study design more clear. 

Reviewer 2 Report

This paper evaluated the effects of oat β-glucan supplementation on the intestinal microbiota and SCFA production in suckling piglets. The authors have determined several parameters, including body weight gain, intestinal morphology, microbiota composition, and SCFA concentrations. However, this paper has some major issues which need to be improved for further consideration.

1. It is noted that the manuscript needs careful editing by someone with expertise in technical editing paying particular attention to English grammar, technical expressions, spelling, and sentence structure.

2. What was the litter size of the selected litter? What was the age of the selected piglets? What was the average BW of piglets? Which basis was considered for selecting 10 piglets per litter? More information should be included in the M&M.

3. Was it controlled housing conditions? What about the temperature and humidity of the rearing house?

4. Did the authors select four piglets per treatment for sampling (L 112-122)? The number of replication is not enough for animal experiments. Did the author conduct a power analysis for sampling?

5. A detailed method should be described for SCFA determination; not as supplemental.

6. The authors only included body weight gain and average daily gain (suppl.); how about other growth performance parameters? Such as daily feed intake, feed intake to body weight gain.

7. How about the diarrhea incidence/mortality during the experiment?

8. Discussion part should further discuss to illustrate the possible mechanisms/functions of β-glucan on gut bacteria composition and SCFAs production rather than a similar explanation of feed or age of animals.

9. What is take away home message from this study? Authors should include take away home messages from this study in conclusion.

10. References should be revised according to the journal format.

Author Response

We thank the reviewer for constructive feedback on the manuscript! The suggested revisions have improved the manuscript!

1. It is noted that the manuscript needs careful editing by someone with expertise in technical editing paying particular attention to English grammar, technical expressions, spelling, and sentence structure.
Author reply: We have sent the manuscript for a professional linguistic review.

2. What was the litter size of the selected litter? What was the age of the selected piglets? What was the average BW of piglets? Which basis was considered for selecting 10 piglets per litter? More information should be included in the M&M.
Author reply: We have provided more details in the method section. Moreover, we have also provided a schematic outline for the experimental design (new figure 1), which we hope clarify the study design further. The litter sizes varied, but we only included sows that gave birth to at least 10 piglets. In case there were more than 10 piglets in the litter, we have a better chance to get balance in birth weight and sex for the CON and BG groups. We have added data on birthweight in the text, whereas the data on bodyweight are presented in figure 2. 

3. Was it controlled housing conditions? What about the temperature and humidity of the rearing house?
Author reply: We have added further information regarding the housing conditions in the method section.

4. Did the authors select four piglets per treatment for sampling (L 112-122)? The number of replication is not enough for animal experiments. Did the author conduct a power analysis for sampling?

Author reply: Sorry if this description was unclear! We included 4 piglets from each litter (two beta glucan (BG) and two control piglets (CON)) for sampling of jugular vein plasma and rectal swabs, where the same piglets where sampled at 4 time points (at 1 – 4 weeks of age). As we included 5 litters, this ended up in 10 piglets with BG supplement and 10 matching control piglets. We have clarified the description in the text as well as by adding a new figure 1 illustrating the study outline. 

5. A detailed method should be described for SCFA determination; not as supplemental.
Author reply: We have updated the text with a reference to the paper where the method was described. We still keep a description of the modifications of the method in the supplement as it is quite long. This is in line how we did in earlier studies using this method. 

6. The authors only included body weight gain and average daily gain (suppl.); how about other growth performance parameters? Such as daily feed intake, feed intake to body weight gain.
Author reply: The BG and CON supplements were only provided during the suckling period, thus daily feed intake of the sow milk was not possible to record. After weaning the piglets did not receive any supplement, and during this period up to nine weeks age we just recorded the body weight and health status but not the feed intake. Moreover, the piglets were kept within the same groups after weaning (as the sow was moved).

7. How about the diarrhea incidence/mortality during the experiment?
Author reply: We have provided info about this in the manuscript. None of the piglets died. None of the piglets got diarrhea that needed treatment during the study, but some pigs had loose stools after weaning. This info have been added in the manuscript.

8. Discussion part should further discuss to illustrate the possible mechanisms/functions of β-glucan on gut bacteria composition and SCFAs production rather than a similar explanation of feed or age of animals.
Author reply: We have further discussed possible mechanisms and functions in the discussion. 

9. What is take away home message from this study? Authors should include take away home messages from this study in conclusion. 
Author reply: We have clarified this in the conclusion. 

10. References should be revised according to the journal format.
Author reply: The references now follow the format according to the instructions from the journal. 

Reviewer 3 Report

The paper is well presented and well documented. English is good, and the experimental design correct. But some points of detail are problematic.

Line 67 It would be useful to specify whether BGs from oat are really involved in stimulating immunity, or is it a specificity of BGs of fungal origin for instance ?

Lines 73-77 this approach to plasma levels of VFAs is very relevant. To support this approach, and given the weakness of the literature in the colon/plasma comparison, it would have been desirable to study the correlation, especially since you have also measured the vfa in the intestine (supplementary table 3). Can you add such correlation analysis in the core text ?

Line 97 this experimental design, even correct statistically thanks to the piglet nested in the sow, is the major weakness of your study because in terms of microbiota the litter effect (even statistically adjusted) is much stronger than the effect of BG.  

Line 108 : Could you mention average quantities ingested of creep feed per group as it is a way to develop fibre digestion during the suckling period and thus can influence VFA production. This can interfere with oat BG supplementation and may explain lack of effect of BG at low levels. This point must be mentionned in the discussion. To test the BG it would not have been necessary to use creep feed.

Line 388 it does seem that numerical differences are observed for propionate and succinate in W1. Was this significant or a trend? perhaps a covariate would have been useful in this case.

Line 415 the dose is probably to low could you compare with studies reporting positive effect of Oat BG ? (if any)

Line 436 'different sources'  is it particularly related to a particular source of BG (oat vs yeast etc...) ?

Author Response

We thank the reviewer for constructive feedback on the manuscript! The suggested revisions have improved the manuscript!

Line 67 It would be useful to specify whether BGs from oat are really involved in stimulating immunity, or is it a specificity of BGs of fungal origin for instance ?
Author reply: We have updated this part of the text and added that the direct immune stimulatory effects is mainly linked to beta-glucans of microbial origin. 

Lines 73-77 this approach to plasma levels of VFAs is very relevant. To support this approach, and given the weakness of the literature in the colon/plasma comparison, it would have been desirable to study the correlation, especially since you have also measured the vfa in the intestine (supplementary table 3). Can you add such correlation analysis in the core text ?
Author reply: We agree that this is a good suggestion and have added this analysis in the core text. 

Line 97 this experimental design, even correct statistically thanks to the piglet nested in the sow, is the major weakness of your study because in terms of microbiota the litter effect (even statistically adjusted) is much stronger than the effect of BG.  
Author reply: We agree that the effect of litter ended up to be stronger than the effects of the BG supplement. We accounted for the effect of litter as much as possible by having a within litter treatment design and by using a nested random effect in the statistical analysis. 

Line 108 : Could you mention average quantities ingested of creep feed per group as it is a way to develop fibre digestion during the suckling period and thus can influence VFA production. This can interfere with oat BG supplementation and may explain lack of effect of BG at low levels. This point must be mentioned in the discussion. To test the BG it would not have been necessary to use creep feed.
Author reply: Unfortunately, we do not have data on how much creep feed the piglets ingested in this study. This would of course have been valuable information to know about their creep feed intake. When we designed the study, we also discussed the option to exclude the creep feed availability for all piglets in the study. In hindsight this would likely showed the effects of the BG supplements more clearly. We have added more text about the potential impact of creep feed in the discussion.

Line 388 it does seem that numerical differences are observed for propionate and succinate in W1. Was this significant or a trend? perhaps a covariate would have been useful in this case.

Author reply: No, it was not significant. We have tried to analyse propionate with a covariate also, although we decided not to use this in the manuscript.

Line 415 the dose is probably to low could you compare with studies reporting positive effect of Oat BG ? (if any)
Author reply: This could be true. The dose of the beta glucan supplement was based on what has been used and demonstrated to show effects on metabolism in human studies. As we used suckling piglets in our study, we did not want to introduce a too high dose. Moreover, as we used a product SweOat bran BG28, containing 28% of soluble BG, We believe that it would have been challenging to increase the dosage to the suckling piglets. The impact of dose is adressed in the discussion section. 

Line 436 'different sources'  is it particularly related to a particular source of BG (oat vs yeast etc...) ?
Author reply: This has been clarified in the text. 

Round 2

Reviewer 1 Report

The manuscript has been improved a lot, and I think it is ready for acceptance. Good luck!

Reviewer 2 Report

The authors considered all of my comments and suggestions. However, the English language should be further improved.